# Efficiently Measuring the Cognitive Ability of LLMs: An Adaptive Testing Perspective

## Abstract

Large language models (LLMs), like ChatGPT, have shown human-level cognitive ability. Benchmarks from various fields (e.g., Literature, Biology and Psychology) are often used to measure LLM's ability and report standard metrics such as accuracy, recall and F1. However, such method for evaluating LLMs can be inefficient and inaccurate from the cognitive science perspective. Inspired by Computerized Adaptive Testing (CAT) used in psychometrics, we propose an adaptive testing framework for LLM evaluation. Rather than using a standard test set and simply reporting accuracy, this approach dynamically adjusts the characteristics of the test questions, such as difficulty, based on the model's performance. This allows for a more accurate estimation of the model's abilities, using fewer questions. More importantly, it allows LLMs to be compared with humans easily, which is essential for NLP models that aim for human-level ability. Our diagnostic reports have found that ChatGPT often behaves like a "careless student", prone to slip and occasionally guessing the questions. We conduct a fine-grained diagnosis and rank 6 commercial instruction-tuned LLMs from three aspects of Subject Knowledge, Mathematical Reasoning, and Programming, where GPT4 can outperform other models significantly and reach the cognitive ability of middle-level students. Different tests for different models using efficient adaptive testing — we believe this will become the new norm in large language model evaluation.

## 1 Introduction

In recent months, large language models (LLMs) have subverted people's perception of NLP model with their powerful capabilities. To fully understand it, an increasing number of researchers have focused their efforts on evaluating its abilities in various aspects. In addition to traditional NLP benchmarks, LLM has shown incredible human-level performance in writing, examination, programming, etc (OpenAI, 2023a). We believe this is just the tip of the iceberg of its latent knowledge.

Recent instruction-tuned LLMs (e.g., ChatGPT) have emerged human-level ability, thus more and more professional and academic exams in various subjects are used to test them, which are originally designed for humans (Figure 1(a)). However, traditional evaluation methods (Qin et al., 2023; Orzechowski & Moore, 2022; Drummond & Japkowicz, 2010; Hernández-Orallo et al., 2021) relying on a fixed exam/benchmark are not efficient for the following reasons: It usually requires many experts in the corresponding domain to score every single response of LLM, especially for the subjective or creative questions. For example, GPT4 official technical report (OpenAI, 2023a) covers more than 30 academic exams, such as History, Literature, Biology and Psychology. Although more evaluations are resorting to crowdsourcing or even LLMs themselves (Li et al., 2023; Törnberg, 2023; Chang et al., 2023), its professionalism, proficiency, and biases are the destabilizing factors. Meanwhile, for today's generative NLP models, the inference overhead can not be negligible. Even for the old GPT3, it needs to generate the response on a 175 billion parameters model token by token. Recent GPT4 limits the frequency of API requests and charges at least 0.03$ for 1K tokens (OpenAI, 2023b), increasing the overhead of evaluation.

To address these issues, we introduce a promising testing method known as Computerized Adaptive Testing (CAT) (Linden et al., 2000), a system widely employed in educational assessment, for the evaluation of LLMs. *CAT's primary goal is to measure examinee's ability accurately while reducing the test length*, which has been widely used in various standardized tests (e.g., GRE and GMAT). It

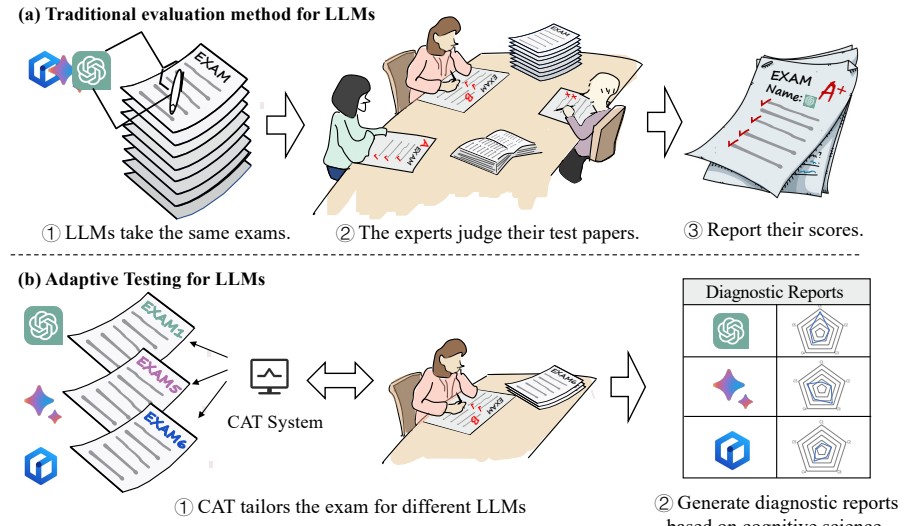

**(a) Traditional evaluation method for LLMs**

① LLMs take the same exams. ② The experts judge their test papers. ③ Report their scores.

**(b) Adaptive Testing for LLMs**

CAT System

Diagnostic Reports

① CAT tailors the exam for different LLMs ② Generate diagnostic reports based on cognitive science.

Figure 1: Traditional evaluation method vs Adaptive testing. (a) LLMs need to answer the same questions, and many experts are required to score their responses. (b) In adaptive testing, CAT can adaptively select *few and best-fitting* questions and generate their diagnostic reports.

is a sequential and iterative framework, using the acclaimed Cognitive Diagnosis Model (e.g., Item Response Theory (IRT) (Embretson & Reise, 2013)) in psychometrics to estimate the current ability of the examinee based on their previous responses. Following this, the adaptive question selection algorithm can pick the next appropriate/valuable items based on specific informativeness metrics (Lord, 2012; Chang & Ying, 1996; Bi et al., 2020), e.g., selecting the one with difficulty closest to his/her current ability estimate. As such, if CAT perceives an underestimate of the examinee's ability, it will opt for a more challenging question in the next step, and vice versa. Compared to traditional paper-and-pencil tests, CAT has been proven to require fewer questions to achieve the same measurement accuracy (i.e., evaluation efficiency) (Lan et al., 2014; Vie et al., 2017).

Our objective is to establish an adaptive and efficient evaluation framework for LLMs. As illustrated in Figure 1(b), we treat LLM as a real student and tailor an "exam" to accurately estimate its ability. Compared to traditional evaluation methods (e.g., fixed benchmarks and case studies (Zhuo et al., 2023; Huang et al., 2023)), it provides us with a scientific solution for measuring the cognitive ability level of LLMs, greatly reducing costs (e.g., labor costs and computational overhead). Our main contributions are as follows:

- We formally introduce CAT into the evaluation of LLMs and propose a practical two-stage adaptive evaluation framework, which enables the efficient comparison between model and model, model and human. Different from the traditional fixed-benchmark evaluation, it requires much fewer questions/samples under the same ability estimation accuracy.

- Model vs Human: We compare ChatGPT with human of different levels: we found that ChatGPT often behaves like a "careless student" who is prone to slip and occasionally guesses questions. Although there is still a gap with high-ability humans, especially in mathematical reasoning, ChatGPT's programming ability in Dynamic Programming and Search has surpassed the high-ability college students.

- Model vs Model: We study 6 instruction-tuned LLMs and provide their fine-grained diagnosis reports on three aspects: subject knowledge, mathematical reasoning, and programming level. Through comparison, it is found that GPT4 surpasses other large models with significant advantages.

## 2 RELATED WORKS

Computerized Adaptive Testing (CAT) is a complex system (Linden et al., 2000), which includes two core algorithms: Item Response Theory (IRT) and question selection algorithm. At each test step

$t \in [1, 2, ..., T]$, these two algorithms work alternately until the stopping rule is met. When the test stops ($t = T$), the estimated ability of individual examinees $\hat{\theta}^T$ will be fed back to themselves for facilitating future learning, or as the basis/result of this assessment. The goal of CAT is to accurately estimate examinee's true ability $\theta_0$, i.e., $\|\hat{\theta}^T - \theta_0\| \to 0$, while minimizing $T$ (i.e., the number of questions asked) (Chang, 2015). The following reviews these two algorithms.

**Item Response Theory.** Item Response Theory (IRT) is built on psychometrics and cognitive science, which is used for ability estimation in several state assessments, such as the National Assessment of Educational Programs (Ravitch, 1995) and OECD/PISA Project (Harlen, 2001). There are many different IRT implementations, the simplest of which is the one-parameter logistic form:

$$\Pr(\text{the response to question } j \text{ is correct}) = sigmoid(\theta - \beta_j). \tag{1}$$

This model represents the behavior of an examinee with a single latent trait $\theta$, called ability, and the questions with a single parameter $\beta$, called difficulty. Note that the characteristics of each question (e.g., difficulty) should be pre-calibrated before CAT by fitting a joint model of human ability and item characteristics to human response patterns to the test questions (Embretson & Reise, 2013). Although more and more neural network-based IRT and cognitive diagnosis models (Wang et al., 2020; 2021; Gao et al., 2021) have been designed recently for ability/proficiency estimation, we choose the IRT in logistic function considering its versatility and interpretability in this paper. With its reliability in model evaluations (Rodriguez et al., 2021), IRT itself has been widely used to evaluate NLP systems, e.g., textual entailment recognition (Lalor et al., 2016), chatbots (Sedoc & Ungar, 2020), and machine translation (Hopkins & May, 2013; Otani et al., 2016).

**Selection Algorithms.** The selection algorithm is the core component to realize CAT's adaptivity – accurately estimating examinee's ability with the fewest test steps. Commonly, these algorithms are based on some uncertainty or information metrics. The most widely used is Fisher Information metric (FSI) (Lord, 2012; Hooker et al., 2009), designed for IRT, which selects the next question that can minimize the uncertainty/variance of estimation. Based on FSI, many improved methods (Chang & Ying, 1996; Rudner, 2002; van der Linden, 1998; Zhuang et al., 2022a) have been proposed to introduce additional information in selection. Recently, Active learning and Reinforcement Learning (RL) are also used to select important/suitable items from the question bank (Bi et al., 2020; Nurakhmetov, 2019; Li et al., 2020; Ghosh & Lan, 2021; Zhuang et al., 2022b). Taking into account both theoretical guarantees and interpretability, the Fisher method is the first choice for the evaluation of LLMs in this paper.

## 3 EVALUATION FRAMEWORK FOR LLMS

In this section, we take ChatGPT as an example to introduce our adaptive evaluation framework for LLMs in detail (Figure 2). Instead of comparing on the unseen gold-standard test dataset, this method can use CAT to (1) realize the comparison of ChatGPT and humans in knowledge level, and (2) use as few samples as possible. To this end, we evaluate it on different educational datasets from three online educational platforms in experiments. They all consist of large-scale students' practice logs on different subjects/domains for human-LLM comparison. In principle, it can be any academic and professional exam (e.g., SAT, Leetcode, and AP exams).

Generally, in the above datasets, given $n$ test questions $Q = \{q_1, ..., q_n\}$ and $m$ examinees (LLM or real human being) $S = \{s_1, ..., s_m\}$, where each examinee answers some questions in $Q$ and gets the binary outcomes $Y = \{0, 1\}$ of correct ($y = 1$) or incorrect ($y = 0$). We can get the response data $D = \{(s_i, q_j, y_{ij}) | s_i \in S, q_j \in Q, y_{ij} \in Y\}$. The detailed two-stage evaluation process is described below.

### 3.1 STAGE 1: CONSTRUCTION OF QUESTION POOLS

A diverse and high-quality question bank is the basis of adaptive testing (Wang & Vispoel, 1998). Before the formal educational assessment for LLM begins, we use the question set $Q$ in the above dataset to construct the question pool (Figure 2): Calibrating the characteristics/parameters of all the questions in $Q$. Thus, an Item Response Theory (IRT) model is fit to the large-scale response data $D$ to obtain such item parameter estimates to support computerized test administration. Previous work

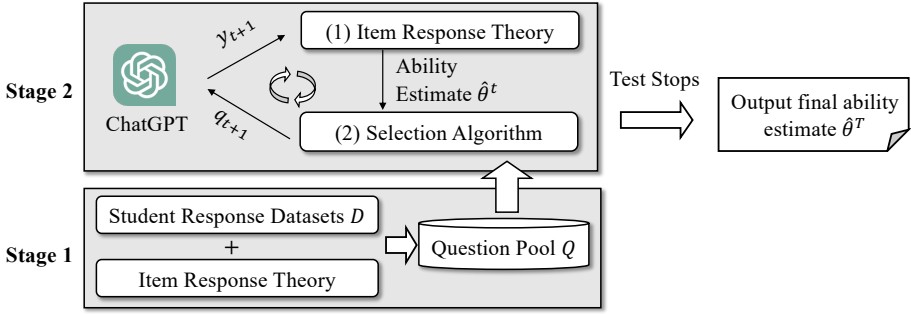

Figure 2: The adaptive testing framework for LLMs.

(Rodriguez et al., 2021) shows that the more sophisticated models are better for evaluating the NLP models, so we adopt the three-parameter logistic (3PL-IRT):

$$p_j(\theta_i) = \Pr(y_{ij} = 1|\theta_i) = c_j + (1 - c_j)\frac{1}{1 + \exp(-\alpha_j(\theta_i - \beta_j))}, \qquad (2)$$

where $p_j(\theta_i)$ is the probability that an examinee $i$ with ability $\theta_i$ gives a correct response to question $j$, and Eq.(2) defines three parameters (difficulty $\beta_j$, discrimination $\alpha_j$, and guessing factor $c_j$) for each question $j$. With the response data $D = \{(s_i, q_j, y_{ij})\}_{i,j}$, joint maximum likelihood estimation can be used to estimate all parameters:

$$\{\alpha_j, \beta_j, c_j\}_{j=1}^{n}, \{\hat{\theta}_i\}_{i=1}^{m} = \arg\max_{\alpha,\beta,c,\theta} \prod_D p_j(\theta_i)^{(y_{ij})}(1 - p_j(\theta_i))^{(1-y_{ij})}, \qquad (3)$$

where $\{\alpha_j, \beta_j, c_j\}_{j=1}^{n}$ are the estimated parameters of all questions, and $\{\hat{\theta}_i\}_{i=1}^{m}$ are the real humans' estimated ability (distribution), which can be used for subsequent LLMs comparisons with humans. Therefore, a dataset that can be used for comparing LLMs with humans needs to contain: (1) response data from real humans and (2) the question's content. In traditional evaluation, to achieve this comparability, human groups and LLMs should answer the same question set or exam, and compare their scores or accuracy. Luckily, IRT only needs each examinee to answer a small part of the whole question pool and does not require them answering the same questions (Lord, 2012).

**Question Characteristics.** In fact, *questions are not equally important for evaluating LLMs*. For example, the two LLMs A and B with an accuracy of 0.88 and 0.89 on one benchmark, their gap may not be as small as it seems. Because, (1) the massive easy samples/questions may overwhelm the difficult ones, so that B cannot show its strong performance over A; (2) or there are annotation errors/noise in the dataset, making the metric fail. IRT's fundamental assumption is that questions are not equal (Lalor et al., 2016). Different questions usually have different characteristics (e.g., difficulty, discrimination, and guessing factors): *(1) Difficulty $\beta$:* The examinee's ability $\theta$ and difficulty $\beta$ have a unified scale. When $\theta$ remains the same, the larger $\beta$ is, the smaller the probability of a correct response. *(2) Discrimination $\alpha$:* For the questions with high $\alpha$, slight changes in ability may lead to large changes of the probability $p(\theta)$, thus these items can better differentiate the examinees with similar abilities. *(3) Guessing factor $c$:* The parameter $c \in [0, 1]$ mainly reflects the probability of low-ability examinees answering the question correctly. As the level is higher, the effect of $c$ becomes smaller. More illustrations and cases about question characteristics can be found in Appendix A.2

### 3.2 STAGE 2: ADAPTIVE TESTING

After the construction of the question pool, the formal CAT starts in a question-LLM interactive mode. In this paper, LLM's latent trait/ability can also be denoted by $\theta$. For accurate and efficient assessment of its true ability $\theta_0$, CAT can sequentially select the best-fitting questions for LLM from the question pool $Q$; then uses its responses for ability estimation. When the test stops, the final estimate is output as the result. To achieve such adaptability, it includes two components: (1) Ability Estimation using IRT and (2) Question Selection, and they work alternately at each test step:

**(1) Ability Estimation using IRT.** For adaptive question selection during testing process, IRT is used to estimate LLM's current ability $\hat{\theta}^t$. Besides, we will illustrate the statistical properties of this

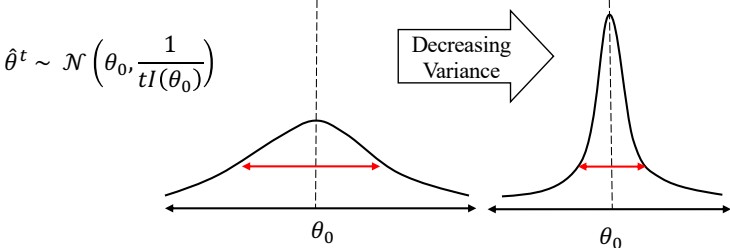

Figure 3: The statistical properties of the ability estimator $\hat{\theta}^t$.

estimate (Figure 3). Specifically, at test step $t \in [1, 2, ..., T]$, given the LLM's previous $t$ responses $S_t = \{(q_1, y_1), ..., (q_t, y_t)\}$, where $\{q_j\}_{i=1}^t \subseteq Q$ are selected sequentially by the selection algorithm and $y$ is the binary outcomes of correct or incorrect; LLM's current ability can be estimated using maximum likelihood estimation (MLE):

$$\hat{\theta}^t = \arg\max_{\theta} \ln \prod_{S_t} p_j(\theta)^{(y_j)} (1 - p_j(\theta))^{(1-y_j)}, \tag{4}$$

where $p_j(\theta)$ represents the probability of the response $(q_j, y_j)$ in IRT, which is defined in Eq.(2). It has been proved that when the sample size $t$ is large, the distribution of estimator $\hat{\theta}^t$ is approximately normal with mean $\theta_0$ and variance $\frac{1}{tI(\theta_0)}$ (Ross, 2014; Efron & Hinkley, 1978) ($I(\theta_0)$ is the Fisher information for $\theta_0$):

**Theorem 1** *(Ross, 2014) Let examinee's responses $(q_1, y_1), ..., (q_t, y_t)$ of size $t$ from a distribution for which the pdf or pmf is $f(\theta) = p_j(\theta)^{(y_j)}(1 - p_j(\theta))^{(1-y_j)}$, with $\theta$ the unknown ability parameter. Assume that the true ability is $\theta_0$, and the MLE result is $\hat{\theta}^t$. Then the probability distribution of $\hat{\theta}^t$ tends to a normal distribution:*

$$\hat{\theta}^t \sim \mathcal{N}\left(\theta_0, \frac{1}{tI(\theta_0)}\right) \tag{5}$$

Obviously, it can be obtained that as the number of test items ($t$) or the Fisher information ($I$) increases, the variance ($\frac{1}{tI(\theta_0)}$) will continue to decrease. As shown in Figure 3, since the estimated value is asymptotically unbiased (i.e., its mean is equal to the true value $\theta_0$), when its variance decreases, the distribution will keep "tightening", thus reducing the uncertainty of the estimated ability $\hat{\theta}^t$. Therefore, increasing $t$ and the Fisher information are the two keys to improving the estimation accuracy.

**(2) Question Selection.** In order to boost the efficiency of ability estimation and reduce the test length $t$, it is crucial to minimize the variance (i.e., maximize $I(\theta_0)$). An important feature of $I(\theta)$ is that the contribution of each question to the total information is additive: $I(\theta) = \sum_{j=1}^t I_j(\theta)$, where $I_j(\theta)$ is Fisher information for question $j$. Therefore, the total amount of information for a test can be readily determined, and we can sequentially select $T$ questions so that their Fisher information at $\hat{\theta}^t, t = 1, 2, ..., T$, are as large as possible. More specifically, it retrieves the next question $q_{t+1}$ from pool $Q$ based on LLM's current estimate $\hat{\theta}^t$:

$$q_{t+1} = \arg\max_{j \in Q} I_j(\hat{\theta}^t), \tag{6}$$

where $I_j(\theta) = \frac{[p'_j(\theta)]^2}{p_j(\theta)[1-p_j(\theta)]}$ can be viewed as the informativeness of question $j$. After receiving new response $y_{t+1}$, IRT will update and estimate ability $\hat{\theta}^{t+1}$ using Eq.(4). Compared with other complex selection algorithms (Chang & Ying, 1996; Bi et al., 2020; Ghosh & Lan, 2021; Zhuang et al., 2022b), this Fisher information method is theoretically guaranteed and more interpretable.

Put the specific IRT formula into $I_j(\theta)$ and we can find that the Fisher method will select questions with (1) high discrimination and (2) difficulty close to the current ability estimate ($\hat{\theta}^t$) (Lord, 2012; Wang & Chang, 2011). Therefore, Fisher method not only considers question's value (i.e., discrimination), but also the adaptability of question's difficulty to the examinee's ability. For example, when

ChatGPT gets it right in step $t$, the algorithm will choose a more difficult question for it, and vice versa. This is why many high-ability GRE examinees in reality find that the test questions become more and more difficult. In Section 4, we compare the efficiency of this adaptive testing framework with the traditional evaluation method.

## 4 DIAGNOSTIC REPORTS FOR LLMS

In this section, we first verify the evaluation efficiency of the proposed adaptive framework. Then, taking ChatGPT as an example, we compare the LLM with humans from three aspects Subject Knowledge (MOOC), Mathematical Reasoning (MATH) and Programming (CODE) (Section 4.2). Finally, we measure the latest 6 instruction-tuned LLMs and rank them by cognitive ability (Section 4.1). The code can be found in `https://anonymous.4open.science/r/CAT4LLM-D6C5`.

**Datasets.** We choose three datasets to conduct fine-grained evaluation of LLM from three key areas: Subject Knowledge Level, Mathematical Reasoning Level, and Programming Level. These datasets are respectively known as MOOC, MATH, and CODE. (1) **Subject Knowledge Level (MOOC)**: Massive Open Online Courses (MOOC) is currently one of the most popular online learning systems, and this dataset[1] collects students' answer records on various knowledge concepts in computer science (e.g., Computer System, Data Structure, and Machine Learning). (2) **Mathematical Reasoning Level (MATH)**: The MATH dataset contains mathematical test items and logs of high school examinations. It covers students from 378 high schools in more than 130 cities. (3) **Programming Level (CODE):** The CODE dataset includes the code submissions of students from more than 120 universities. It is collected from an online programming platform. Due to anonymity principle, we omit the name of MATH and CODE datasets. Appendix A.1 shows the statistics of the datasets.

**Experimental Setup** First, as mentioned in Section 3.1, all examinee response data in the three datasets should be used to estimate the question parameters (Eq.(3)) for constructing the question pools. It is worth mentioning that each dataset needs to be divided into a validation set to prevent overfitting. Second, the CAT system interacts with LLM for multiple rounds: LLM answers the questions selected by the selection algorithm, then IRT updates the ability estimate based on this response. Since the response from LLM is relatively lengthy, especially when answering fill-in-the-blank or short-answer questions, an automated method is not practical and an expert is required to judge its correctness. Such LLM-CAT-Expert interactions are shown in Appendix A.3.

**Compared Examinees.** In this paper, in addition to the popular ChatGPT, we compare human student with 6 commercial instruction-tuned LLMs: **High/Mid-Ability Student** (It refers to the ability value of the Top 20%/50% of all students in the datasets), **ChatGPT** (OpenAI), **GPT4** (OpenAI), **Bard** (Google), **ERNIEBot** (Baidu), **QianWen** (Alibaba), and **Spark** (iFlytek).

### 4.1 COMPARISON OF DIFFERENT LLMS

In addition to ChatGPT, we also use the above CAT method to compare the cognitive level of other models (Table 1). More importantly, in order to intuitively compare the abilities with humans, we also show the ability estimates of high-ability (Top 20%) and middle-ability (Top 50%) students, where CODE and MOOC are college students, and MATH is high school students.

**GPT4 is the Best.** GPT4 is significantly higher than other LLMs in terms of mathematical reasoning, programming, and subject knowledge level. In particular, the subject level of GPT4 surpasses high-ability college students (Top 20%) in almost every knowledge concept. A large amount of knowledge can be "stored" with its massive training data and unprecedented model parameters, which is one of the reasons why other language models cannot beat it.

**Each LLM has its own strengths.** For example, for programming level (CODE), GPT4 is good at Dynamic Programming and Math Problem, and ChatGPT is good at Search Problem. Although Spark's average programming ability is lower than that of GPT4, using programming to solve

---

[1]https://www.biendata.xyz/competition/chaindream_mooccube_task2/

Table 1: Estimated value ($\hat{\theta}$) for students and each model. The boldfaced indicates the highest ability value among these LLMs. The underline "_" indicates that the model surpasses mid-ability students (Top 50%). "*" indicates this model surpasses high-ability students (Top 20%).

| | | Instruction Tuned LLMs | | | | | | Student Top | |
|---|---|---|---|---|---|---|---|---|---|
| | | Bard | ChatGPT | GPT4 | ERNIEBOT | QianWen | Spark | 20% | 50% |
| MATH | Equations and Inequalities | 0.55 | 0.44 | ***0.77** | 0.46 | 0.37 | *0.66 | 0.65 | 0.55 |
| | Probability and Statistics | 0.36 | 0.14 | **0.59** | 0.14 | 0.14 | 0.37 | 0.66 | 0.57 |
| | Function | 0.36 | 0.48 | 0.49 | 0.26 | 0.14 | **0.58** | 0.65 | 0.55 |
| | Permutation and Combination | 0.12 | 0.03 | **0.58** | 0.25 | 0.13 | 0.57 | 0.65 | 0.56 |
| | Geometry | 0.22 | 0.01 | 0.35 | **0.36** | 0.24 | 0.25 | 0.66 | 0.56 |
| | *Average* | 0.32 | 0.22 | 0.56 | 0.29 | 0.21 | 0.49 | 0.65 | 0.56 |
| | *Rank* | *High-Ability > GPT4 ≈ Mid-Ability > Spark > Bard > ERNIEBOT > ChatGPT > QianWen* | | | | | | | |
| CODE | Dynamic Programming | 0.34 | *0.72 | ***0.83** | 0.42 | 0.42 | 0.40 | 0.70 | 0.63 |
| | Data Structure | 0.37 | 0.40 | **0.40** | 0.29 | 0.29 | 0.29 | 0.67 | 0.58 |
| | Math Problem | 0.46 | 0.60 | ***0.84** | 0.39 | 0.39 | 0.60 | 0.66 | 0.58 |
| | Search | 0.23 | ***0.73** | 0.51 | 0.41 | 0.41 | 0.41 | 0.70 | 0.61 |
| | Tree and Graph Theory | 0.00 | 0.38 | **0.49** | 0.27 | 0.34 | 0.27 | 0.63 | 0.54 |
| | *Average* | 0.28 | 0.57 | 0.61 | 0.35 | 0.37 | 0.40 | 0.67 | 0.59 |
| | *Rank* | *High-Ability > GPT4 > Mid-Ability > ChatGPT > Spark > ERNIEBOT > QianWen > Bard* | | | | | | | |
| MOOC | Programming Language | ***0.80** | 0.57 | *0.78 | 0.26 | 0.47 | 0.57 | 0.73 | 0.63 |
| | Machine Learning | *0.78 | *0.67 | ***0.99** | *0.77 | *0.88 | 0.25 | 0.55 | 0.48 |
| | Computer System | 0.68 | 0.70 | ***0.82** | 0.49 | 0.38 | 0.48 | 0.74 | 0.66 |
| | Data Structure | 0.66 | **0.67** | 0.66 | 0.23 | 0.03 | 0.56 | 0.69 | 0.60 |
| | Algorithm | ***1.00** | *0.79 | *0.77 | 0.34 | 0.46 | 0.43 | 0.69 | 0.60 |
| | *Average* | 0.78 | 0.68 | 0.80 | 0.42 | 0.44 | 0.46 | 0.68 | 0.60 |
| | *Rank* | *GPT4 > Bard > ChatGPT ≈ High-Ability > Mid-Ability > Spark > QianWen > ERNIEBOT* | | | | | | | |

mathematical problems is its forte. Therefore, although many LLMs have not announced the specific details of the data used, we have reason to infer that e.g., ChatGPT/GPT4 uses more coding-related data, and Spark uses more mathematics-related data in the training stage.

**Mathematical reasoning of LLM still has a long way to go.** Mathematical reasoning ability is an important aspect for evaluating LLMs. Unfortunately, according to the estimated ability output by CAT, even the well-performing GPT4 and Spark models are only equivalent to mid-ability high school students. After all, the essence of LLM is still the sequence-to-sequence generative model based on probability instead of thinking and reasoning like humans. Transformer obviously is not enough to imitate human cognitive structure or process. Therefore, *problem-solving based on cognition/reasoning (Liu et al., 2023; Ding et al., 2019; Lin et al., 2021) is still lacking in LLMs.*

**Evaluation Efficiency.** In addition to the theoretical guarantees, we use simulation experiments to verify the evaluation efficiency of the framework: Due to the unknown of the true ability $\theta_0$, we artificially generate 100 examinees' $\theta_0$ and conduct the Simulation of Ability Estimation experiment on the MATH dataset using the mean square error $\mathbb{E}[\|\theta^t - \theta_0\|^2]$ between the ability estimate $\theta^t$ at each step and the true ability $\theta_0$ (Figure 4(a)): Fisher method can reduce the evaluation error quickly. Compared with using a fixed test set (randomly sampled from the data distribution), *such adaptive evaluation method in this paper only needs 20% of the questions at most under the same estimation accuracy.* Therefore, especially for tests that require human experts to score, this solution can greatly reduce labor costs and improve the efficiency of LLMs' evaluation. As 20 is sufficient for the length of a typical adaptive test, we fix the max length to 20 and adaptively adjust the test length according to the informativeness metric (Wang et al., 2018). Therefore, rather than evaluating on hundreds of questions (OpenAI, 2023a; Huang et al., 2023), adaptive testing method can pick out truly valuable questions for evaluation, and only need a maximum of 20 questions.

**Adaptive Question Selection.** To determine whether Computerized Adaptive Testing can adaptively select appropriate questions based on a model's ability, we employ the Jaccard similarity coefficient to measure the similarity between the test questions answered by any two models. This is defined as $Jaccard(A, B) = |A \cap B|/|A \cup B|$, where $A$ and $B$ represent two different question sets. Figure 4(b) shows the Jaccard similarity of the test questions selected by CAT for each LLM (on MATH). Remarkably, almost all Jaccard values hover around 0.6, indicating that at least 20-30% of the questions are distinct, which is crucial for achieving the adaptivity of testing. In addition, the remaining 70-80% of the questions in these exams answered by the LLMs are the same, and

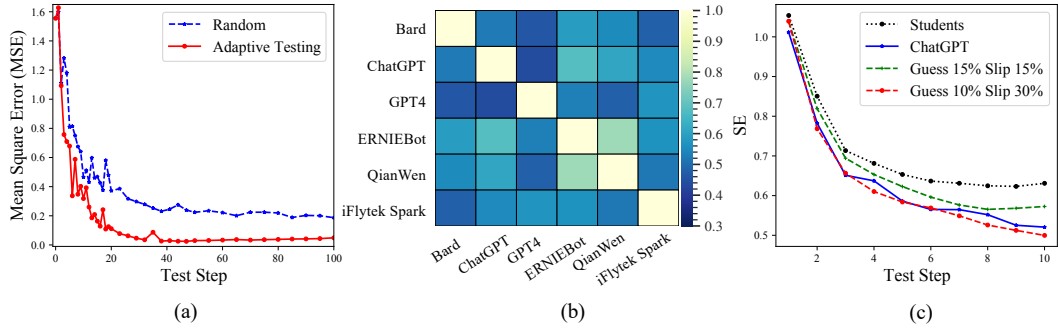

Figure 4: (a) Simulation experiments of ability estimation using MSE: $\mathbb{E}[\|\hat{\theta}^t - \theta_0\|^2]$. (b) The average Jaccard similarity coefficient of the selected questions for each LLM. (c) SE curves of ChatGPT and students with different guess and slip factors during adaptive testing.

are valuable for evaluating all LLMs. Together, these two segments compose a test paper that can effectively evaluate the model and enhance the precision of ability assessment.

**Adaptive Testing's Reliability: ChatGPT is a "Careless Student".** To confirm whether the adaptive testing framework used for humans can be used for LLMs, we study its reliability (SE curve (Wang et al., 2018; Choi et al., 2011)). In the context of CAT, the SE value often refers to the standard error of ability estimate $\hat{\theta}^t$, which reflects the precision of an examinee's ability estimate: $SE(\hat{\theta}^t) = 1/\sqrt{\sum_{j=1}^t I_j(\hat{\theta}^t)}$. A smaller SE indicates a more precise or reliable estimate (Van der Linden & Glas, 2010; Wang et al., 2018). Figure 4(c) shows the SE changes during the testing process of ChatGPT (blue) and 100 students (black). Although ChatGPT's SE curve is not stable, it is faster and easier to converge than the student. To investigate the characteristics of ChatGPT SE curve and gain deeper insights on its similarity with humans, we add the guess and slip factors (Zhuang et al., 2022b)) to the student's testing process: (1) Guess factor: even if examinee doesn't master the question, there is a small chance of answering it correctly; (2) Slip factor: when encountering a simple one, there may be a small chance to answer it wrong. Thus, Guess10% means that the correctness label changes from 0 to 1 with 10%, and Slip10% means that the true label has a 10% probability of changing from 1 to 0. Interestingly, ChatGPT's SE curve is very close to the student SE curve of Guess=10%, Slip=30% (red). From this, we can deduce that ChatGPT behaves like a "careless student" who is prone to slip (30%) and occasionally guesses the answers (10%).

## 4.2 CHATGPT VS HUMAN

In this part, we take ChatGPT as an example to evaluate it as a real human, using this adaptive testing framework. First, we compare ChatGPT and high-ability humans from three aspects, and provide a fine-grained diagnostic report. Next, we investigate the reliability of the CAT framework for LLM, and further explore the similarity between humans and LLM. Many other findings can be found in Appendix.

**(1) Subject Knowledge Level:** Figure 5 shows the ability comparison between ChatGPT and real students. In Figure 5(a), the ability level of ChatGPT in the two concepts of Algorithm and Machine Learning is significantly higher than that of high-ability students. The programming language is the weakest part of ChatGPT, which obviously does not match his superior performance in coding ability as illustrated in (Kashefi & Mukerji, 2023; Biswas, 2023). To explore the reason, the right shows a very basic question case about Programming Language, but ChatGPT gets it wrong. Obviously, it is not proficient in grasping and understanding some basic concepts in programming languages. Combined with its amazing coding level on CODE (Figure 5(c)), we have reason to believe: ChatGPT is more like a "doer" rather than a "nerd".

**(2) Mathematical Reasoning Level:** From Figure 5(b), there is still a considerable gap between the mathematical reasoning ability of ChatGPT and that of humans. Surprisingly, during the test, ChatGPT incorrectly answers almost all questions about Probability and Statistics, Permutation and Combination, and Geometry. But its performance on Functions, Equations and Inequalities is

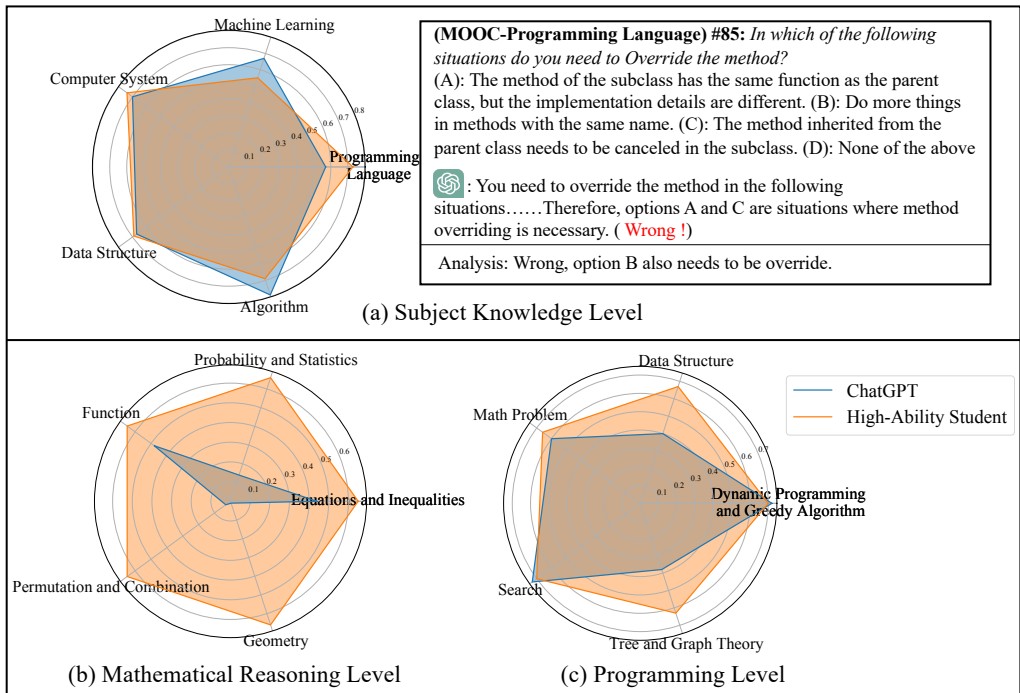

Figure 5: The diagnostic report (i.e., the normalized final ability estimate $\hat{\theta}^T$ on different concepts) of ChatGPT on three aspects.

relatively much better. Therefore, for such basic calculation problems with fixed problem-solving routines, ChatGPT is still competent. However, ChatGPT does not have the ability to solve the questions that require reasoning from real-world scenarios (Lin et al., 2023) (e.g., Probability and Statistics, Permutation and Combination).

**(3) Programming Level:** Although ChatGPT has shown its amazing coding capabilities both in the official reports and enormous user cases, it is not omnipotent nor good at all types. We use the CODE programming platform to conduct a fine-grained evaluation of ChatGPT's programming ability (Figure 5(c)), including Dynamic Programming and Greedy Algorithm, Search, Math Problem, Data structure, and Tree and Graph Theory. The strongest are Search, Dynamic Programming and Greedy Algorithm, which can greatly surpass high-ability college students. However, Data Structure, and Tree and Graph Theory are its shortcomings. Therefore, next time you ask ChatGPT to write code, please try to avoid these types, and if you encounter problems about dynamic programming, please feel free to hand it over to ChatGPT.

## 5 CONCLUSION AND FURTHER WORKS

More and more users are trying to explore LLM's abilities in different aspects, and even ask it to do some things that "normal" NLP models cannot do, such as generating code, making PowerPoint, and writing emails. Thus, how to scientifically and efficiently evaluate its ability is more and more important. In this paper, we leverage an adaptive testing framework for assessing humans: Computerized Adaptive Testing (CAT). With its high efficiency, fewer questions are required under the same evaluation accuracy, which greatly reduces the labor cost and computation overhead.

This paper is the initial attempt at evaluating LLMs using adaptive testing. Obviously, in terms of technology of the paper, it is very simple yet interpretable. Item Response Theory is unidimensional, while more complex cognitive science models, such as cognitive diagnosis, can be considered for a multidimensional and comprehensive assessment of the model.

Furthermore, in addition to the ability estimation for LLM in this paper, some important concerns with LLMs, such as hallucinations, unfairness, security, and robustness, can also be estimated by designing corresponding selection algorithms to enhance assessment efficiency.

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

## A    APPENDIX

### A.1    STATISTICS OF THE DATASETS

**Datasets.**    We choose three datasets to conduct fine-grained evaluation of LLM from three key areas: Subject Knowledge Level, Mathematical Reasoning Level, and Programming Level. These datasets are respectively known as MOOC, MATH, and CODE. Table 2 show the statistics of the datasets.

- Subject Knowledge Level (MOOC): Massive Open Online Courses (MOOC) is currently one of the most popular online learning systems, and this dataset[2] collects students' answer records on various knowledge concepts in computer science (e.g., Computer System, Data Structure, and Machine Learning).

---

[2]https://www.biendata.xyz/competition/chaindream_mooccube_task2/

Table 2: Statistics of the datasets.

| Dataset | #Examinees | #Questions | #Response logs | Concept (#Questions) |
|---|---|---|---|---|
| MOOC | College Students (15866) | 592 | 66437 | Computer System(132), Programming Language(155), Data Structure(100), Algorithm(93), Machine Learning(38) |
| MATH | High School Students (107674) | 2242 | 176155 | Probability and Statistics(61), Permutation and Combination(73), Geometry(190), Function(328), Equations and Inequalities(105) |
| CODE | College Students (1388) | 207 | 7913 | Dynamic Programming and Greedy Algorithm(26), Search(26), Math Problem(37), Data Structure(42), Tree and Graph Theory(13) |

- Mathematical Reasoning Level (MATH): The MATH dataset is collected from a widely-used online learning platform, which contains mathematical test items and logs of high school examinations. It covers students from 378 high schools in more than 130 cities.

- Programming Level (CODE): The CODE dataset includes a large number of code submissions of students from more than 120 universities. It is collected from an online programming platform.

## A.2 DETAILED EXPLANATION ABOUT QUESTION CHARACTERISTICS

In fact, *questions are not equally important for evaluating LLMs*. For example, the two LLMs A and B with an accuracy of 0.88 and 0.89 on one benchmark, their gap may not be as small as it seems. Because, (1) the massive easy samples/questions may overwhelm the difficult ones, so that B cannot show its strong performance over A; (2) or there are annotation errors/noise in the dataset, making the metric fail. IRT's fundamental assumption is that questions are not equal (Lalor et al., 2016). The following illustrate the questions of different difficulty, discrimination, and guessing factors:

*(1) Difficulty $\beta$:* The examinee's ability $\theta$ and difficulty $\beta$ have a unified scale. When $\theta$ remains the same, the larger $\beta$ is, the smaller the probability of a correct response. The following are the hardest and easiest questions in the MOOC dataset. Solving problem #9 (left) requires ChatGPT to understand PAC, a very difficult concept in computational learning theory. In contrast, the question about the basic "perceptron" with the smallest $\beta$ (right) can be solved easily.

---

*High Difficulty ($\beta = 0.967$)*
**Q#9**: *What two conditions should a PAC-learnable learner meet?*
(A): The learner must output a hypothesis with an arbitrarily high probability of an arbitrarily low error rate. (B): Good stability. (C): The time of the learning process grows in a polynomial manner at most. (D): There are more data features and the number of samples is sufficient  (MOOC-Machine Learning)

**Response from ChatGPT:** The correct answers are A and C. In order to be PAC-learnable, a learning algorithm must satisfy two conditions. The first ... The second condition ... While stability and the number of data features may be desirable properties for certain applications, they are not requirements for a PAC-learnable algorithm. (Correct !)

---

*Low Difficulty ($\beta = 0.133$)*
**Q#13**: *Is the perceptron a generative model or a discriminative model?*
(A): Generative model.
(B): Discriminative model.  (MOOC-Machine Learning)

**Response from ChatGPT:** The perceptron belongs to the discriminative model category, which is option B. In machine learning, ... thus belongs to the discriminative model category. (Correct !)

---

*(2) Discrimination $\alpha$:* For the questions with high $\alpha$, slight changes in ability may lead to large changes of the probability $p(\theta)$, thus these items can better differentiate the examinees with similar abilities. We find that the question #23 (with low discrimination) is very straightforward and there is a fixed routine for such "perpendicular bisector" questions, which is inappropriate to differentiate different levels of ability. The highly discriminative question #169 (left) is not difficult but not as straightforward as #23. A simple transformation of it is required and utilize the concept of "Positional Relationship between Circles", but obviously ChatGPT has not fully grasped them.

*High Discrimination ($\alpha = 2.103$)*
**Q#169**: *In the Cartesian coordinate system $xOy$, given the circle $C : (x - a)^2 + (y - a + 2)^2 = 1$, the point $A(0,2)$, if there is a point $M$ on the circle $C$, satisfying $MA^2 + MO^2 = 10$, then the value range of the real number $a$ is {blank}.*
(MATH-Geometry)

**Response from ChatGPT:** In the Cartesian coordinate system $xOy$, given circle $C$ and point $A(0,2)$. If there exists a point $M$ …then the range of real numbers for a is $[1 - \sqrt{6}, 1 + \sqrt{6}]$. To solve this problem, we can use the distance formula to find the distance between points. Let $O$ be the center of circle $C$, which is $(a, a - 2)$. Then, we have: $MA^2 = (X_M - 0)^2 + (y_M - 2)^2$, $MO^2 = (x_M - a)^2 + (y_M - a + 2)^2$. Substituting ... (Wrong !)

*Low Discrimination ($\alpha = -2.753$)*
**Q#23**: *Given the points $A(1,2), B(3,1)$, then the equation of the perpendicular bisector of line segment AB is:*
(A): $4x + 2y = 5$.  (B): $4x - 2y = 5$.
(C): $x + 2y = 5$.  (D): $x - 2y = 5$.
(MATH-Geometry)

**Response from ChatGPT:** To find the equation of the perpendicular bisector of segment AB, we need to find its midpoint and slope. The midpoint can be found by taking the average of the $x$-coordinates and the average of the $y$-coordinates: Midpoint $= \left(\frac{1+3}{2}, \frac{2+1}{2}\right) = (2, 1.5)$. The slope ... Therefore, the equation of the perpendicular bisector of segment AB is option B: $4x - 2y = 5$. (Correct !)

*(3) Guessing factor $c$:* The parameter $c \in [0, 1]$ mainly reflects the probability of low-ability examinees answering the question correctly. As the level is higher, the effect of $c$ becomes smaller. For the question #81 with $c = 0.951$, it doesn't even require the examinee to master any knowledge about any programming language, and can be "guessed" correctly from common sense. To answer the question #74 (right) with the smallest $c$, ChatGPT needs to master the definition and usage of Interface in JAVA: "An interface is a completely abstract class", and be able to pick out all irrelevant options; It is almost impossible to guess correctly without fully grasping the concept of "Interface".

*High Guessing Factor ($c = 0.951$)*
**Q#81**: *Among the following statements about parent classes and subclasses, the correct one is*
(A): A subclass can have multiple parent classes. (B): A parent class can have multiple subclasses. (C): A parent class can only have one subclass. (D): None of the above answers are correct.
(MOOC-Programming Language)

**Response from ChatGPT:** Regarding the statements about parent and child classes, the correct statement is B: a parent class can have multiple child classes. This is a fundamental concept in object-oriented programming, ... (Correct !)

*Low Guessing Factor ($c = 0.054$)*
**Q#74**: *In Java, the keywords that cannot be used to modify the keyword "interface" are*
(A): private.  (B): public
(C): final  (D): static
(MOOC-Programming Language)

**Response from ChatGPT:** The correct answers are A: private and D: static. In Java, interfaces are implicitly public and abstract, so the access modifier "public" is redundant when used with an interface. The "final" keyword is also allowed for use with interface methods and fields, but not with the interface itself. However, the "private" keyword is not allowed for ... (Wrong !)

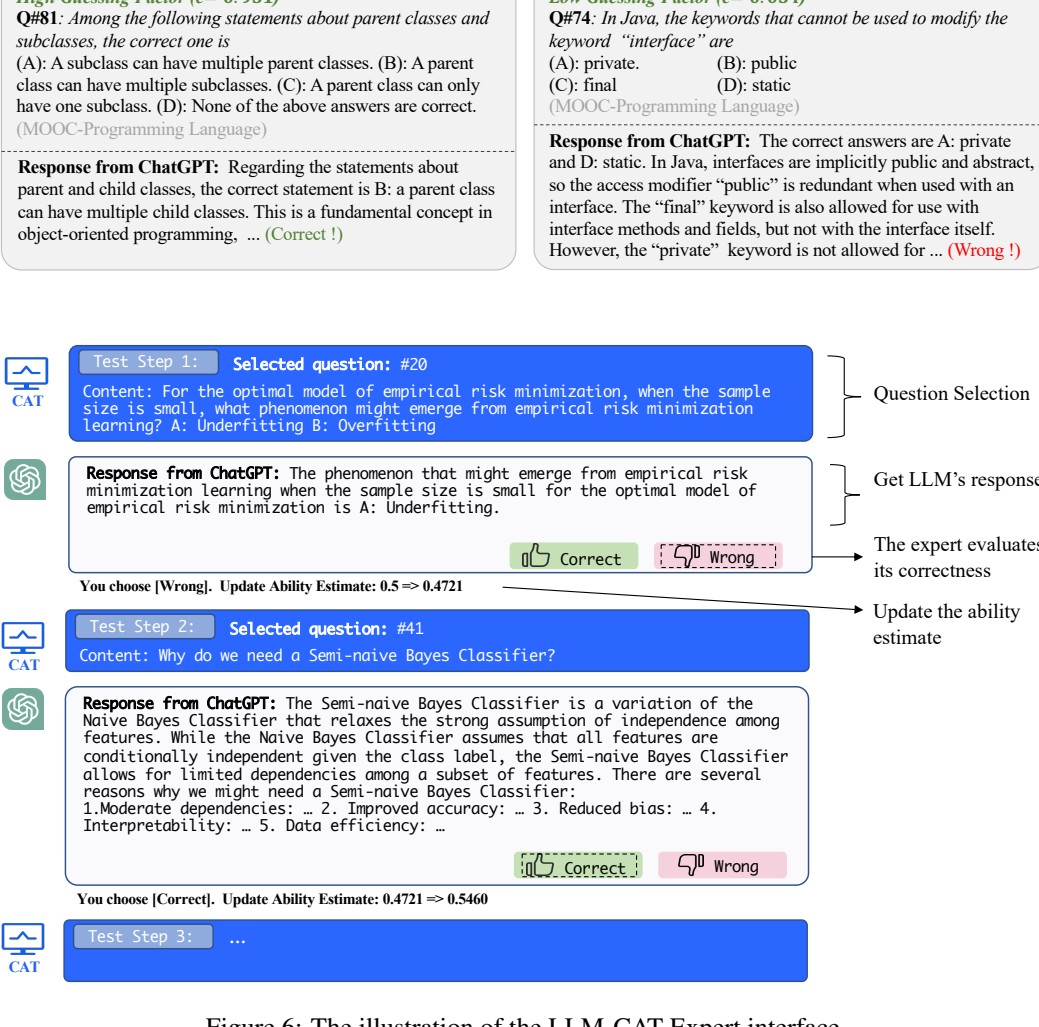

Figure 6: The illustration of the LLM-CAT-Expert interface

## A.3   LLM-CAT-Expert interface

Such LLM-CAT-Expert interactions are shown in Figure 6. It will automatically add the following prompt to ask LLM before each question: *You are an examinee and please answer the following question: [content of the question].*

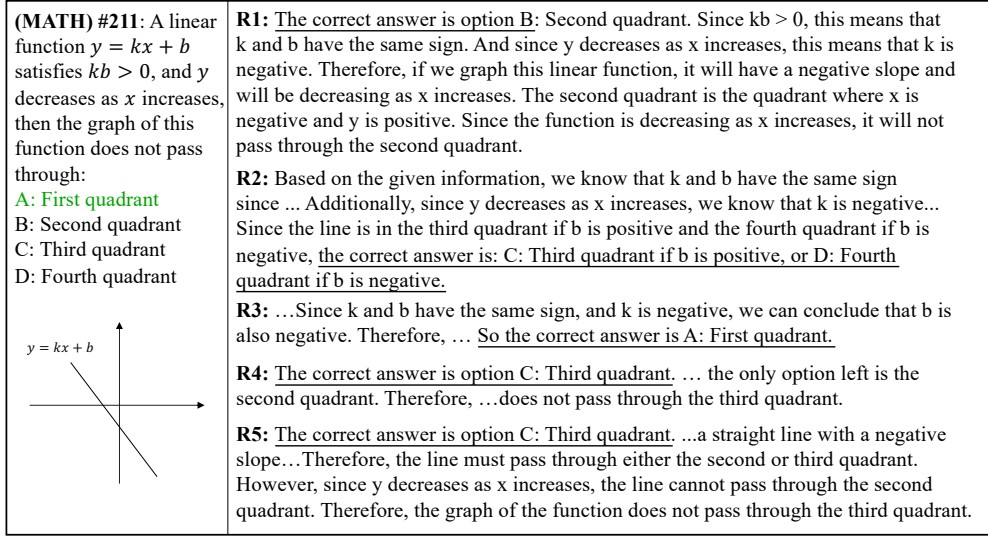

Figure 7: The illustration of ChatGPT's "Fickle-Minded" characteristic: it answers the same question 5 times, and gives 4 different answers (only R3 is correct).

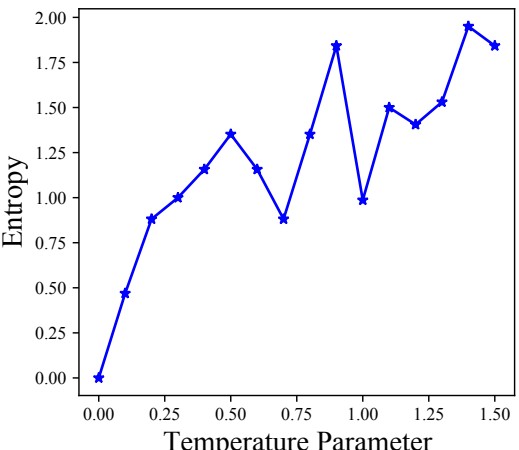

Figure 8: Response Uncertainty vs Temperature Parameter of ChatGPT.

## A.4   ChatGPT is "Fickle-Minded"

In the testing of ChatGPT, we discover that if it is asked to answer the same question repeatedly, it often produces varied and even contrasting responses. Figure 7 illustrates that it provides four different answers to the same question asked five times in different sessions. This "inconsistency" is largely due to ChatGPT being a probability-based generative model; while this ensures each response is diverse and creative, it also introduces potential issues. As a result, this inconsistency creates noise/uncertainty during the test. We also investigate the impact of the temperature parameter, which controls the level of randomness or creativity in the text generated by ChatGPT (OpenAI, 2023b). Figure 8(b) shows that as the temperature increases, so does the uncertainty of ChatGPT's answers. Therefore, when asking the ChatGPT to solve rigorous problems (such as mathematics), a lower temperature parameter is preferred.

