# OpenReview forum: "Efficiently Measuring the Cognitive Ability of LLMs: An Adaptive Testing Perspective"
_ICLR.cc/2024/Conference — Submitted to ICLR 2024_

### Official Review · Reviewer_AbmR · 2023-10-31

**Soundness:** 3 good
**Presentation:** 4 excellent
**Contribution:** 3 good
**Rating:** 5
**Confidence:** 4

**Summary:**

The paper proposes to leverage computerized adaptive testing for evaluating language models. Merits of doing so include 1) fewer questions to accurately evaluate models' capabilities and 2) making them comparable between models and between models and humans. The evaluation procedure involves two iterative processes: 1) estimating "skill" using IRT, and 2) recommend the next question based on some uncertainty measure (Fisher information). The paper uses the proposed method to evaluate a number of commercially available models (and humans) on 3 datasets, and makes various interesting observations about models' behaviors, which are also interpretable thanks to the interpretability of the inferred parameters in IRT.

**Strengths:**

The idea of using CAT for evaluating LLMs appears novel. My takeaway from this new evaluation method is that
1. we can use a lot fewer probing questions to accurately evaluate LLMs ability on a specific subject, given a pool of questions and a lot of students' answers to calibrate IRT parameters, instead of on the entire pool of questions, which may become useful in resource constraint settings, and
2. we can interpret and characterize LLMs using the inferred IRT "ability", "forget" etc. parameters

**Weaknesses:**

I don't have much complaint about this paper besides a few clarity questions which I put in the "questions" section. Those questions center around the IRT parameter estimation, the calibration process, comparability between humans and models, and interpreting Figure 4c. Due to these questions, I put my rating as 5 for now but will update the score after the authors clarify.

**Questions:**

1. How does the "human response" calibration in IRT work? Does it mean that the question difficulty parameter is pre-estimated using students' responses, and is then fixed during the subsequent CAT procedures?
2. Following up the previous question, what IRT parameters are fixed and are estimated at each estimation step $t$ during CAT? My impression from the paper is that $\theta$ is being estimated, $\beta$ is fixed (or pre-estimated or calibrated), but what about the remaining parameters?
3. Is the calibration process done independently for each dataset, or the responses from all three datasets are combined to estimate $\beta$?
4. The calibration process is done using human data, and the calibrated IRT model is used to evaluate LLMs. I think the underlying assumption is that LLMs behave like humans and thus can be evaluated using human-data calibrated IRT model. I am not sure if such assumption is valid, or can be verified; can you comment on this? Because if not, it does not make sense to use human-calibrated IRT model to evaluate LLMs because they are not comparable. Does it make more sense to have LLMs answer the questions, use these answers (in addition to human ones) to calibrate the IRT model, and then evaluate?
5. It seems (from Appendix A.3) that some of the questions in the MATH data contain figures. How are models evaluated when the questions contain both texts and figures? My impression is that all models being evaluated are text-only models.
6. In figure 4c, what does the student curve mean? Is it one student, or the average of many students (The caption says "students" whereas the text says "student")? The student curve appears to be more uncertain; does it imply the student(s) is/are (on average?) more careless than the model?

---

> ### Author Response · Authors · 2023-11-14
> **Rebuttal by Authors**
>
> We would like to express our gratitude for your appreciation of the contribution and novelty presented in our paper. Your feedback means a lot to us. Here are the responses to each of your questions:
>
> > **Q1**: How does the "human response" calibration in IRT work? Does it mean that the question difficulty parameter is pre-estimated using students' responses, and is then fixed during the subsequent CAT procedures?
>
> **A1**: Yes. Just like the general CAT method, we can use human response to estimate the question parameters. For example, we can use the BCE loss on the binary responses data to estimate the parameters, and fix them for direct use in the subsequent CAT process.
>
> > **Q2**: Following up the previous question, what IRT parameters are fixed and are estimated at each estimation step $t$ during CAT?My impression from the paper is that $\theta$ is being estimated, $\beta$ is fixed (or pre-estimated or calibrated), but what about the remaining parameters?
>
> **A2**: All parameters (except ability $\theta$) need to be pre-estimated. All the parameters of question are fixed (e.g., difficulty $\beta$, discrimination $\alpha$, guess factor $c$). The only one that can update continuously during the CAT process is the ability estimate $\theta$.
>
> > **Q3**: Is the calibration process done independently for each dataset, or the responses from all three datasets are combined to estimate $\beta$?
>
> **A3**: Yes, the difficulty of questions in different datasets/domains cannot be aligned, so parameter estimation are independent for each dataset.
>
> > **Q4**: The calibration process is done using human data, and the calibrated IRT model is used to evaluate LLMs. I think the underlying assumption is that LLMs behave like humans and thus can be evaluated using human-data calibrated IRT model. I am not sure if such assumption is valid, or can be verified; can you comment on this? Because if not, it does not make sense to use human-calibrated IRT model to evaluate LLMs because they are not comparable. Does it make more sense to have LLMs answer the questions, use these answers (in addition to human ones) to calibrate the IRT model, and then evaluate?
>
> **A4**: This is a very good question. We use students' data to estimate parameters and apply them to the LLM, because we want to achieve human and LLM comparability. "Cognitive Ability" is actually viewed from a human perspective, even if ChatGPT can answer global challenges correctly, but fails to answer simple addition and subtraction, we still think it is problematic in cognitive ability. If we want to achieve a simple ranking between models, of course we can also use model response data to estimate parameters, or even use a pre-trained difficulty predictor to predict the characteristics of any question in the world. This depends on the purpose of evaluating the LLM.
>
> > **Q5**: It seems (from Appendix A.3) that some of the questions in the MATH data contain figures. How are models evaluated when the questions contain both texts and figures? My impression is that all models being evaluated are text-only models.
>
> **A5**:  Sorry for causing you misunderstanding. The figure in Appendix A.3 do not exist in the dataset. For readers to better understand, we drew this picture manually to illustrate fickle-minded issue of ChatGPT.
>
> > **Q6**: In figure 4c, what does the student curve mean? Is it one student, or the average of many students (The caption says "students" whereas the text says "student")? The student curve appears to be more uncertain; does it imply the student(s) is/are (on average?) more careless than the model?
>
> **A6**: This figure represents the estimated uncertainty averaged over many students. A smaller value indicates a more precise or reliable ability estimate [1] and this is not directly related to the degree of carelessness. We find that after adding various types of perturbations (guess and slip) the curve will be lower and the estimate will be more reliable. The reason may be that the parameters in IRT (e.g., guess and discrimination factor) can directly model these perturbations. Actually, by comparing the positions of four curves, the ChatGPT is more careless than the normal students.
>
> Reference:
>
> [1] Elements of adaptive testing[M]. New York: Springer, 2010.

---

### Official Review · Reviewer_YD4n · 2023-11-01

**Soundness:** 3 good
**Presentation:** 3 good
**Contribution:** 3 good
**Rating:** 5
**Confidence:** 3

**Summary:**

The standard way to evaluate LLMs is to use metrics such as accuracy and use all of the samples to evaluate. However, this may be inefficient and costly, especially under API frequency limits and charges. This paper use ideas from computerized adaptive testing from the education field to evaluate LLMs. The method adaptively selects questions based on the estimate of the ability of the examinee (LLM). The benefit is that we require much fewer questions and samples compared to the traditional way. With the proposed framework, the paper compares various LLMs and humans with MOOC, MATH and CODE datasets.

**Strengths:**

- Evaluation of LLMs is an important topic.
- Recently, many of the tools and studies traditionally used to evaluate humans are being applied to assess LLMs, but to my knowledge, an evaluation method inspired by Computerized Adaptive Testing (CAT) is novel.
- The paper is well organized.
- The proposed framework is applied to model-model comparison and model-human comparison.
- Code is provided.

**Weaknesses:**

- The emphasis on evaluation efficiency was not immediately evident to me. While I recognize that inference can carry significant costs, it's relatively more economical compared to the (pre-)training of LLMs. Furthermore, evaluations are typically conducted once, rendering them comparatively affordable. I can also imagine LLM providers would want to evaluate their LLMs very accurately, to demonstrate that their LLM is better than others with high confidence. In contrast to the fact that humans become tired after answering too many questions, LLM performance will not degrade as we ask more questions. Given these considerations, it would be helpful if the paper can provide more motivating situations where efficiency can be important.
- While the comparative analysis of the abilities of various LLMs (such as lack of mathematical reasoning, being a careless student, GPT-4 is the best, etc.) is interesting, it would be beneficial to delineate specifically which aspects of the findings represent novel insights (that was not shown in previous papers). This would enhance the clarity and depth of the discussion.

**Questions:**

- What is the ChatGPT model? Is it gpt-3.5-turbo?

---

> ### Author Response · Authors · 2023-11-14
> **Rebuttal by Authors**
>
> Thank you for your valuable feedback on our paper. We appreciate the time and effort you have invested in reviewing our work. We understand that there might be room for improvement in terms of clarity and depth.
>
> [The necessity of evaluation efficiency.]
>
> - **Save the cost of model inference**. Simply put, it saves money. It is difficult to estimate the time and money costs incurred by model inference on such a large scale as GPT3.5. We can calculate the cost of using the API. Take WMT benchmark in machine translation as an example. It has a total of 11 million tokens. Considering the API price of ChatGPT, a complete evaluation of one model costs 20$. This seems acceptable, but if we want to evaluate and rank 10 models on 10 different domain benchmarks, it may not be. Recently, more and more comprehensive large benchmarks, various LLMs (general/specific to a certain domain) have emerged, and the iteration speed of versions of the same model is also accelerating. In order to achieve fast, timely and accurate assessment, CAT is a good choice (obviously not the only).
>
> - **Save the cost of human experts**: In order to automatically evaluate the correctness of model responses, most benchmarks can only include multiple-choice questions. For free response questions (e.g., subjective, creative questions) that truly test the model's generation capabilities, human experts' judgments are more reliable than GPT-4 and ordinary people (on crowdsourcing platforms) as stated in the paper. However, the energy of human experts is limited (humans may become tired as you said), we need to find truly valuable questions for LLM to answer and let the experts judge their responses.
>
> - Last but not least, as mentioned in the paper, evaluation methods based on cognitive science such as CAT can be used for building a scientific evaluation paradigm, which can achieve comparisons between models and humans, and between models.
>
> > **Q**: What is the ChatGPT model? Is it gpt-3.5-turbo?
>
> **A**: Yes. We will include more LLMs for adaptive evaluations in future work.
>
>
> We truly appreciate your suggestion about the presentation aspect of the motivation and experiments. Please feel free to share any further insights or suggestions you might have.

---

> > ### Comment · Reviewer_YD4n · 2023-11-22
> > **Comment**
> >
> > Thank you for providing answers to my comments and questions. The 2nd example raised (saving the cost of human experts) about free response questions without a multiple-choice answer is interesting, and seems like a potentially nice application. However, the datasets used in the paper seem to have fixed answers (at least for the examples shown in Appendix A.2, it seemed like this is the case.) It may strengthen the motivation of the paper by adding open ended datasets in future versions of the paper. Taking this into account, I would like to raise my score for "Contribution" to "3 good". Overall, While I appreciate the effort to address the concerns about the motivation, I still feel the strength of the motivation can be improved, and I am inclined to maintaining my "Rating".

---

> > > ### Author Response · Authors · 2023-11-22
> > > **Rebuttal by Authors**
> > >
> > > Thank you for your thoughtful feedback and for considering raising your score for "Contribution".
> > >
> > > I understand your point about the strength of the motivation. In the revised version of the paper, I will make an effort to better articulate the motivation and, as you suggested, I will consider incorporating open-ended datasets to further strengthen the applicability of our work.
> > >
> > > Again, thank you for your valuable comments. Please feel free to share any further insights or suggestions you might have.

---

### Official Review · Reviewer_Mwhi · 2023-11-02

**Soundness:** 2 fair
**Presentation:** 3 good
**Contribution:** 3 good
**Rating:** 3
**Confidence:** 4

**Summary:**

This paper proposes a new evaluation framework that can dynamically adjust the characteristics of the test questions based on the model's performance. It claims to offer ease for LMs to compare with humans, and the dynamic evaluation pipeline is more accurate in evaluating the model's abilities.

**Strengths:**

**Clarity and Significance**

The presentation of the work is clear and easy to follow. The evaluation is extensive and manages to demonstrate the claimed advantage of the proposed method.

**Weaknesses:**

My concerns about this work are two-fold: 1) As far as I know, one main/initial advantage of using CAT for evaluation is to save human effort during evaluation in human studies. For example, to evaluate the student's academic performance (e.g., GRE, which is the example you used in the paper), we can use CAT to shorten the length/time duration of the evaluation. I wonder why this is the case for LLM evaluation --- now the candidates are machines rather than humans, so I guess we don't care that much about "human effort" or "labor cost" here (we can easily shorten the evaluation time by running models in parallel)?

Even though it does have the above-mentioned advantage, this actually raises another concern: if every model has 20%-30% difference in their evaluation questions (as mentioned in your paper), how to make sure the evaluation is fair to each model? Do you calibrate the final score by the difficulty level or any other factors? Even if this method can produce more accurate results, I don't think it will offer ease for LLMs to compare with each other (e.g., in a leaderboard), since they are probably based on different test sets which may add a lot of communication costs. Calibration might be able to mitigate this but I believe it will introduce more noise into the evaluation pipeline.

In general, I agree with the authors that the CAT is a promising method for future evaluation (e.g., for human education), but some advantages claimed by the authors might not be valid in the context of LLM.

**Questions:**

Are there any other baseline methods? In Table 1 you show some rankings in terms of CAT evaluation results. What if you evaluate your models with the original static test sets of these tasks? Will you arrive at the same rankings? If not, what could be the reason?

---

> ### Author Response · Authors · 2023-11-14
> **Rebuttal by Authors**
>
> Thank you for your valuable feedback. Regarding the questions you raised, we have carefully considered each point and have made following responses:
>
> > **Concern 1**: The necessity of evaluation efficiency.
>
> - **Save the cost of model inference**. Simply put, it saves money. It is difficult to estimate the time and money costs incurred by model inference on such a large scale as GPT3.5. We can calculate the cost of using the API. Take WMT benchmark in machine translation as an example. It has a total of 11 million tokens. Considering the API price of ChatGPT, a complete evaluation of one model costs 20$. This seems acceptable, but if we want to evaluate and rank 10 models on 10 different domain benchmarks, it may not be. Recently, more and more comprehensive large benchmarks, various LLMs (general/specific to a certain domain) have emerged, and the iteration speed of versions of the same model is also accelerating. In order to achieve fast, timely and accurate assessment, CAT is a good choice (obviously not the only).
>
> - **Save the cost of human experts**: In order to automatically evaluate the correctness of model responses, most benchmarks can only include multiple-choice questions. For free response questions (e.g., subjective, creative questions) that truly test the model's generation capabilities, human experts' judgments are more reliable than GPT-4 and ordinary people (on crowdsourcing platforms) as stated in the paper. However, the energy of human experts is limited, we need to find truly valuable questions for LLM to answer and let the experts judge their responses.
>
> - Last but not least, as mentioned in the paper, evaluation methods based on cognitive science such as CAT can be used for **building a scientific evaluation paradigm**, which can achieve comparisons between models and humans, and between models.
>
> > **Concern 2**: Fairness about evaluation.
>
> We deeply appreciate your thoughtful consideration of the fairness issues associated with CAT itself. Like recommendation system (different users will be recommended different items), CAT is also a personalized question in Education and has fairness issues of course e.g., [1], [2], which is another interesting topic. The typical application of CAT is GRE exam: when a test-taker begins the GRE, the computer provides a question of medium difficulty. If the response is correct, the computer presents a slightly more challenging question. Conversely, if the response is incorrect, the following question will be slightly easier. That's why some candidates may perceive the test becoming progressively more easier and their final score will be relatively low. The final score or ranking is related to the accuracy rate and the difficulty of the questions (answered by him). The CAT tests of each candidates are independent, and there are no communication costs. As mentioned in the paper, in Stage 1, we can collect some responses to estimate and fix the question parameters. Then, any model (that may do not appear in Stage 1) can successfully proceed to Stage 2 testing and evaluation. In an improved version, we will give a brief overview of fairness issues in CAT itself.
>
> > **Q1**: Are there any other baseline methods?
>
> **A1**: In this work, No. Because this paper is the first attempt to apply CAT to the assessment of LLM. We abandon the complex methods and technical implementations in CAT and choose the simplest Fisher selection method, which is the most interpretable among them. We would like to introduce the CAT idea to more LLM researchers. We are trying to introduce the CAT idea to more LLM researchers and ICLR readers in a friendly way.
>
>
> > **Q2**: In Table 1 you show some rankings in terms of CAT evaluation results. What if you evaluate your models with the original static test sets of these tasks? Will you arrive at the same rankings? If not, what could be the reason?
>
> **A2**: Sure. In fact, we compared two leaderboards generated using the full dataset (containing at least thousands of samples) and 20 questions selected by CAT. We found that the consistency between these two leaderboards can reach 100%. We did not add this results to the paper because the number of candidate LLMs is small, making this ranking unreliable. On the other hand, as stated in the paper, all questions in the question bank/dataset are not equally important. Some low-quality questions (or noise data) will interfere with the evaluation and ranking. Filtering out these data is what CAT is trying to do. Therefore, we resort to the metric in traditional CAT field, such as simulation experiments (Figure 4a) and estimation uncertainty analysis (Figure 4c).
>
>
>
>
> [1] Martin A J, Lazendic G. Computer-adaptive testing: Implications for students’ achievement, motivation, engagement, and subjective test experience[J]. Journal of educational psychology, 2018, 110(1): 27.
>
> [2] Flaugher R L. The many definitions of test bias[J]. American Psychologist, 1978, 33(7): 671.

---

### Official Review · Reviewer_E6Gc · 2023-11-09

**Soundness:** 1 poor
**Presentation:** 1 poor
**Contribution:** 1 poor
**Rating:** 3
**Confidence:** 2

**Summary:**

The paper uses adaptive testing, which dynamically adjusts subsequent components of assessment according to performance of the assessee, to assess several LLMs' ability in " Subject Knowledge, Mathematical Reasoning, and Programming".

**Strengths:**

Employing CAT to quickly converge to an accurate skill assessment sounds like a sensible way to improve efficiency, if efficiency is indeed a problem in assessment. I don't think the domain the authors have identified actually has a real evaluation problem, though, and it's not clear to me that CAT would generalize to other domains that are much more expensive, such as free response questions.

**Weaknesses:**

- I can't tell what the core motivation is, or how this approach addresses it. I think that it's efficiency, but I can't tell why in these assessment domains, efficiency is actually a problem.

  For example, what are "destabilizing factors" like "professionalism, proficiency, and biases"? What is being destabilized, exactly?

  Or, if we are assessing Subject Knowledge, can't that be done by MCQA, which doesn't require expert annotation once the benchmark is created? If we are assessing programming, can't we check that with pre-specified unit tests? Indeed, the appendix shows that the unidentifiable datasets used in this paper are multiple choice, as opposed to free response. In these cases, how many forward passes through the model are we actually saved by CAT, and is that substantial? I can't tell from the paper.  But I don't think it works to motivate a paper by saying "free response is expensive so we propose this efficient method and evaluate it on multiple choice"

  What "specific informativeness metric" will we use to choose subsequent "valuable items" to ask the model?
- I also am not sure I understand how one would propose to evaluate whether CAT can be used for LMs. It's not clear to someone for whom CAT is a new idea exactly how item response theory (which is "built on cognitive science") could generalize to LLMs when LLM "cognition" does not seem to match humans'. This paper seems to skip this part of the argument and just use CAT for LMs
- false or at the very least fringe claims asserted without evidence or even argument
	- "Large language models (LLMs), like ChatGPT, have shown human-level cognitive
	  ability"
	- "We believe this is just the tip of the iceberg of its latent knowledge."
- sloppy writing (grammatical mistakes, ambiguous sentences/claims, )

**Questions:**

Where in the paper do you explain how much efficiency we get from using this method, if this is indeed, the motivation? Doesn't the efficiency argument only hold in free response questions, as opposed to MCQA, as you've evaluated on here?

**Details Of Ethics Concerns:**

I am very worried about evaluating the abilities' of LMs with unsubstantiated evaluations, built, in this case, for evaluating human performance. I'm not sure if I am misunderstanding the nature of CAT, and I may be, but this seems like a problematic substantiation to skip.

---

> ### Author Response · Authors · 2023-11-13
> **Rebuttal by Authors**
>
> Thank you for your thoughtful comments and questions regarding our manuscript. Please feel free to share any further insights or suggestions you might have.
>
> First of all, I need to clarify that CAT does not require whether the model/human responses are multiple choice or free response questions. CAT can run successfully as long as the correctness of the response can be judged by some methods. In this paper we only consider two types: multiple choice and fill-in-the-blank questions. Multiple-choice questions can be automatically judged, while the fill-in-the-blank are judged for correctness by human experts.
>
> [The necessity of evaluation efficiency]:
> - **Save the cost of model inference**. Simply put, it saves money. It is difficult to estimate the time and money costs incurred by model inference on such a large scale as GPT3.5. We can calculate the cost of using the API. Take WMT benchmark in machine translation as an example. It has a total of 11 million tokens. Considering the API price of ChatGPT, a complete evaluation of one model costs 20$. This seems acceptable, but if we want to evaluate and rank 10 models on 10 different domain benchmarks, it may not be. Recently, more and more comprehensive large benchmarks, various LLMs (general/specific to a certain domain) have emerged, and the iteration speed of versions of the same model is also accelerating. In order to achieve fast, timely and accurate assessment, CAT is a good choice (obviously not the only).
>
> - **Save the cost of human experts**: In order to automatically evaluate the correctness of model responses, most benchmarks can only include multiple-choice questions. For free response questions (e.g., subjective, creative questions) that truly test the model's generation capabilities, human experts' judgments are more reliable than GPT-4 and ordinary people (on crowdsourcing platforms) as stated in the paper. However, the energy of human experts is limited, we need to find truly valuable questions for LLM to answer and let the experts judge their responses.
>
> - Last but not least, as mentioned in the paper, evaluation methods based on cognitive science such as CAT can be used for **building a scientific evaluation paradigm**, which can achieve comparisons between models and humans, and between models.
>
>
> > **Q1**: I also am not sure I understand how one would propose to evaluate whether CAT can be used for LMs.
>
> **A1**: Actually, we do not skip this part. On page 8 ("Adaptive Testing's Reliability: To confirm whether the adaptive testing framework used for humans can be used for LLMs, ..."), we experimentally verify that the uncertainty of the model's ability assessment can be converged to the same as that of humans, and even faster.
>
> > **Q2**: false or at the very least fringe claims asserted without evidence or even argument: "Large language models (LLMs), like ChatGPT, have shown human-level cognitive ability","We believe this is just the tip of the iceberg of its latent knowledge."
>
> **A2**: LLMs like ChatGPT, have shown human-level cognitive. How to evaluate a person's "human-level" cognitive ability? We usually conduct a series of tests on him, and the same for the LLM. OpenAI's official report has already reported that ChatGPT has achieved amazing results in various human subject examinations, and we quote it in the first paragraph; "We believe this is just the tip of the iceberg of its latent knowledge.": This is what we believe after reading so many works related to LLM's various abilities. The LLM is constantly developing and improving, and this is our opinion in this paper.
>
> > **Q3**: Where in the paper do you explain how much efficiency we get from using this method, if this is indeed, the motivation?
>
> **A3**: Page 8 of the paper (Evaluation Efficiency) explains the accuracy comparison between random selection (traditional benchmark) and adaptive testing. We can conclude that for a given test length or sample size limit, the CAT method is much better than random selection in terms of ability estimation accuracy.

---

> > ### Comment · Reviewer_E6Gc · 2023-11-21
> >
> > Thanks for clarifying the above points. I felt that your questions to my answers above re-asserted the claims made in the paper rather than bringing new evidence, pointing me to arguments in the paper that I missed, or other steps that would have changed my view on the strength of the paper. Therefore, I don't plan to change my scores at this point.

---

> > > ### Author Response · Authors · 2023-11-22
> > > **Rebuttal by Authors**
> > >
> > > I understand and respect your decision to maintain your current scores. I appreciate your time and effort in reviewing our work.  I would like to reiterate that my responses to your questions are grounded and **already exist in the original paper not re-asserted**.
> > >
> > > - The necessity of evaluation efficiency:
> > >   - Save the cost of model inference ("for today’s generative NLP models, the inference overhead can not be negligible..." on Page 1)
> > >   - Save the cost of human experts (" It usually requires many
> > > experts in the corresponding domain to score every single response of LLM,..." on Page 1)
> > >   - building a scientific evaluation paradigm (", it provides us with a scientific solution for measuring the cognitive ability level of LLMs, greatly reducing costs ..." on Page 2).
> > >
> > > - Q1: Page 8 ("Adaptive Testing's Reliability: To confirm whether the adaptive testing framework used for humans can be used for LLMs, ...")
> > >
> > > - Q2: Page 1 ("In addition to traditional NLP benchmarks, LLM has shown incredible human-level performance in writing, examination, programming, etc (OpenAI, 2023a)."). This reference (OpenAI, 2023a) has clearly states that "GPT-4 exhibits human-level performance on various professional and academic benchmarks" and "These models (LLMs) have the potential to significantly impact society in numerous ways [ref]". Thus I don't think there's any problem with these two sentences.
> > >
> > > - Q3: Page 8 of the paper (Evaluation Efficiency).
> > >
> > > Thank you once again. Please feel free to continue the discussion if you have any further questions.

---

### Author Response · Authors · 2023-11-14
**Official Comment**

Many reviewers question the motivation of the paper (i.e., the necessity of evaluation efficiency). We explain this in the second paragraph of the paper, but now it seems insufficient. Below is the detailed explanation for everyone to check.

[The necessity of evaluation efficiency]:

- **Save the cost of model inference**. Simply put, it saves money. It is difficult to estimate the time and money costs incurred by model inference on such a large scale as GPT3.5. We can calculate the cost of using the API. Take WMT benchmark in machine translation as an example. It has a total of 11 million tokens. Considering the API price of ChatGPT, a complete evaluation of one model costs 20$. This seems acceptable, but if we want to evaluate and rank 10 models on 10 different domain benchmarks, it may not be. Recently, more and more comprehensive large benchmarks, various LLMs (general/specific to a certain domain) have emerged, and the iteration speed of versions of the same model is also accelerating. In order to achieve fast, timely and accurate assessment, CAT is a good choice (obviously not the only), which can select a "valuable" sample subset from full dataset for evaluations.

- **Save the cost of human experts**: In order to automatically evaluate the correctness of model responses, most benchmarks can only include multiple-choice questions. For free response questions (e.g., subjective, creative questions) that truly test the model's generation capabilities, human experts' judgments are more reliable than GPT-4 and ordinary people (on crowdsourcing platforms) as stated in the paper. However, the energy of human experts is limited, we need to find truly valuable questions for LLM to answer and let the experts judge their responses.

- Last but not least, as mentioned in the paper, evaluation methods based on cognitive science such as CAT can be used for **building a scientific evaluation paradigm**, which can achieve comparisons between models and humans, and between models.

We will elaborate further on this in the main text. We thank all reviewers for their valuable insights and suggestions, which is crucial to improve the depth of the paper.

---

### Meta-Review · Area_Chair_Yfzf · 2023-12-14

**Metareview:**

This work aims to evaluate and benchmark LLMs from a different perspective. Instead of building a fixed set of test prompts, it borrows the idea from Computerized Adaptive Testing (CAT) used in psychometrics to develop an adaptive LLM evaluation framework. Specifically, it dynamically adjusts the characteristics of the test questions, such as difficulty, based on the model's performance.

The authors argue this approach effectively measures the cognitive capabilities of LLM. They conclude that GPT-4 exhibits that of middle-level students while GPT-3.5 resembles a ''careless student'' in its performance. From a technical standpoint, the authors argue that this method helps save computing cost and human annotation effort compared to existing LLM evaluation frameworks.

Regrettably, the reviewers have found difficulties in identifying substantial technical contributions within this work. Additionally, they express skepticism regarding the claims made about 1) effectively measuring the cognitive abilities of the LLMs, and 2) the purported benefits of reducing both computational and human annotation efforts for evaluating LLMs. Given these considerations, it is difficult to endorse this submission favorably in its present state.

**Justification For Why Not Higher Score:**

This submission received ratings of 3, 3, 5, and 5, consistently falling short of the borderline. More critically, the main technical contribution made by the authors fail to convince both the reviewers and the AC. On a personal note, I appreciate the concept proposed in this research, as it attempt to address the ongoing challenge of effectively evaluating LLMs—a task not adequately served by existing leaderboards. Nonetheless, both the execution of this idea and the narrative framework used to present the idea of measuring cognitive ability necessitate substantial enhancements prior to consideration for acceptance.

**Justification For Why Not Lower Score:**

n/a

---

### Decision · Program_Chairs · 2024-01-16

Reject